# People's Perception of Climate Change Impacts on Subtropical Climatic Region: A Case Study of Upper Indus, Pakistan

**Bashir Ahmad [1], Muhammad Umar Nadeem [1,2,\*], Saddam Hussain [3,4,\*], Abid Hussain [5], Zeeshan Tahir Virik [6], Khalid Jamil [7], Nelufar Raza [1], Ali Kamran [1] and Salar Saeed Dogar [8]**

[1] Climate, Energy and Water Research Institute, National Agriculture Research Center, Islamabad 44000, Pakistan; directorcewri@parc.gov.pk (B.A.); nelufar-raza@hotmail.com (N.R.); engr.alikamran87@gmail.com (A.K.)

[2] Department of Engineering Mechanics and Energy, Graduate School of Science and Technology, University of Tsukuba, Tsukuba 305-8577, Ibaraki, Japan

[3] Department of Agricultural and Biological Engineering, Tropical Research and Education Center (TREC), University of Florida, Homestead, FL 33031, USA

[4] Department of Irrigation and Drainage, University of Agriculture Faisalabad, Faisalabad 38000, Pakistan

[5] International Centre for Integrated Mountain Development (ICIMOD), Kathmandu 44700, Nepal; abid.hussain@icimod.org

[6] Water, Energy and Environmental Engineering Unit, Faculty of Technology, University of Oulu, 90570 Oulu, Finland; zeeshan.virk@oulu.fi

[7] Water System and Global Change Group (WSG), Wageningen University, 6700 AA Wageningen, The Netherlands; muhammadkhalid.jamil@wur.nl

[8] Agrosphere (IBG-3), Institute of Bio and Geosciences, Forschungszentrum Jülich University, 52428 Jülich, Germany; s.dogar@fz-juelich.de

\* Correspondence: engr.umarthe88@gmail.com (M.U.N.); s.hussain@ufl.edu (S.H.)

**Abstract:** In developing countries like Pakistan, the preservation of the environment, as well as people's economies, agriculture, and way of life, are believed to be hampered by climate change. Understanding how people perceive climate change and its signs is essential for creating a variety of adaptation solutions. In this study, we aim to bridge the gap in current research within this area, which predominantly relies on satellite data, by integrating qualitative assessments of people's perceptions of climate change, thereby providing valuable ground-based observations of climate variability and its impacts on local communities. Field-based data were collected at different altitudes (upstream (US), midstream (MS), and downstream (DS)) of the Upper Indus Basin using both quantitative and qualitative assessments in 2017. The result shows that these altitudes are highly variable in many contexts: socioeconomic indicators of education, agriculture, income, women empowerment, health, access to basic resources, and livelihood diversifications are highly variable in the Indus Basin. The inhabitants of the Indus Basin perceive the climate changing around them and report impacts of this change as increase in overall temperatures (US 96.9%, MS 97%, DS 93.6%) and erratic rainfall patterns (US 44.1%, MS 73.3%, DS 51.0%) resulting in increased water availability for crops (US 38.6%, MS 39.7%, DS 54.8%) but also increasing number of dry days (US 56.7%, MS 85.5%, DS 67.1%). Communities at these altitudes said that agriculture was their primary source of income, making them particularly vulnerable to the effects of climate change and the dangers that go along with it. The insights are useful for determining what information and actions are required to support local climate-related hazard management in subtropical climate regions. Moreover, it is vital to launch a campaign to raise awareness of potential hazards, as well as to provide training and an early warning system.

**Keywords:** climate change; people perception; livelihood adaptations; hazard management; Upper Indus

## 1. Introduction

One of the major issues of the current time is the changing climate [1,2]. According to scientific research, global climate change is increasing the likelihood of extreme climatic events (such as droughts and floods) and the stacking of events (for instance, heavy rain causing landslides and soil erosion) [3]. Moreover, the Intergovernmental Panel on Climate Change (IPCC) [4,5] concluded that due to an increase in evapotranspiration and a decrease in rainfall, droughts will dominate every part of the world, especially in arid and semiarid subtropical regions. Pakistan is among the top ten climate change-vulnerable countries [6,7], where climate change is severely affecting the socioeconomic conditions and food security of small watershed communities, particularly in hill ecology [8]. Majority of remote watersheds face declining water availability, high population growth rate, increasing risk of extreme events (floods/drought/heat wave), changing weather patterns, unaffordable energy costs, and degrading natural resources, which negatively affects the productivity and livelihood of farming communities [9,10]. Currently, most of the existing traditional watershed management practices are not capable of coping with the rapidly emerging risks [11]. Furthermore, the lack of resources, skilled human resources, demonstration sites, and awareness negatively affects community resilience in responding to climate change risks [10].

To successfully drive adaptation and mitigation measures, it is necessary to understand how members of the public relate to climate change through their experiences and how they perceive the implications of climate change [12]. Worldwide, people's perspectives and opinions about climate change are evolving. People are intimately experiencing the effects of climate change on the unique environments in which they live and work due to extreme occurrences like flooding and slow, persistent, chronic events like drought [13]. Residents have a deep connection with nature and have developed an instinctive sense of their surroundings over time [14]. Local perceptions of climate change are influenced by daily interactions with the environment and reliance on weather conditions for survival. Furthermore, compared to model-based perceptions, those who live in situ are better able to cope with the local climate and the effects of extreme weather [15]. Therefore, the thoughts of local inhabitants related to changing climate should be monitored and taken into account in the decision-making process of policies, climate risk management, and adaptation strategies.

The Indus Basin is a significant water resource of Pakistan and helps maintain the natural ecology in the watershed [16]. The majority of people who are dependent on agriculture for their livelihood are adversely impacted by the changing climate. Water bodies in the Indus Basin are under tremendous stress as a result of rising industrialization and population [17]. Poor water management practices increase problems like salinity and water logging, resulting in reduced soil quality and poor agricultural output. Water supplies in the Indus Basin have also been reduced because of rising water pollution, which makes some water sources unfit for particular purposes [18].

Indus Basin's water supplies have come under immense pressure because of the expanding population and increasing development. In Pakistan, total annual water extractions have levitated from 153.4 km$^3$ in 1975 to 183.5 km$^3$ in 2008, whereas total annual renewable water resources per capita have dived from 3385 cubic meters (m$^3$) in 1977 to 1396 m$^3$ in 2011. The depleting water table is another problem occurring due to the unsustainable extraction of groundwater. Dropping water tables pushes farmers to irrigate with even more saline water, salinizing the soils and limiting their production potential. Salt-affected soils now trouble 4.5 million hectares, making up 22 percent of Pakistan's irrigated lands (Qureshi et al., 2010) [19–22].

Studies about the assessment of temperature trends from 1961 to 2000 revealed a consistent increase in the diurnal temperature range across all seasons and significant rises in winter mean and maximum temperatures in the Upper Indus Basin. The projected increase in the median annual air temperature is anticipated to range from 0.8 to 5.7 °C by the end of the 21st century. Research on precipitation in the Indus Basin has revealed

significant variability. In the Upper Indus Basin (UIB), a notable decrease in spring precipitation was observed, with the maximum decline recorded at 5.3 mm/year. The annual precipitation in the Lower Indus Basin (LIB) displayed an increasing trend, with a similar pattern observed in autumn [22–25].

Along with the available scientific data about climatic conditions in these areas, it is important to comprehend how these climatic changes affect the people on the ground [15]. The literature review revealed that there is currently insufficient information available regarding how climatic patterns are perceived by the people living in the Indus Basin and the impacts they experience. Therefore, this detailed study is designed to bridge this gap and examine the perceptions of local communities of the Indus Basin regarding climate change and its associated risks. This research aims to integrate both quantitative and qualitative examinations of the dynamics of socioeconomic parameters, individuals' perceptions of climate change [21–23], their reactions, and the actions taken to adapt to it. For this study, three regions in the Indus Basin were considered based on their location along the river basin. Each region has diverse topographic and climatic conditions [19]. A total of 400 households participated in an app-based survey that operationalized the quantitative analysis. The livelihood seasonal monitoring calendar, semi-structured interviews, and/or focus group discussions (FGD) were all combined in the qualitative assessment. Within participants and a subset of the 12 research areas of the qualitative evaluation, the quantitative assessments were completed. A stratified sampling approach was used to survey the sampled households. The survey, conducted in 2017, solicited responses from individuals regarding their perceptions of various climate indicators and impacts over the past decade. The findings of this study will serve as a foundational basis for ground-level knowledge. Policymakers and other stakeholders can utilize this information to develop climate change adaptation strategies and mitigate the vulnerability of agricultural communities to climate change impacts. By emphasizing the socioeconomic and socio-cultural characteristics of residents in response to changing climate conditions, targeted strategies can be developed.

## 2. Materials and Methods

### 2.1. Study Area

One of Asia's most important waterways is the Indus River. It begins in the Kailash Range in Tibet, flows to the west, and eventually empties into the Arabian Sea (Douglas, 2006) [20]. The Indus River has a length of 2900 km and a drainage area of 966,000 km$^2$. The foundation of Pakistan's surface water resources is the river and its tributaries. The Jhelum, Chenab, Ravi, Sutlej, and Beas Rivers to the east and the Kabul River to the west comprise the Indus River's tributaries. Rainfall in the catchment areas, snowmelt, and glacier runoff account for the majority of these rivers' input (Qureshi, 2011) [21].

To conduct this study about people's perspectives on climate change, the study area of Indus Basin is divided into three primary sites: upstream (Indus Basin, Hunza, and Nager Districts); midstream (Soan Basin); and downstream (Chaj Doab, primarily Sargodha district). There is a great deal of variability in these three regions across a wide range of settings [2,8,20]. The Indus Basin has a wide range of socioeconomic indices in health, access to essential resources, women's empowerment, agriculture, education, and income. The salient features of the chosen districts to investigate people's perceived climate change are shown in Figure 1.

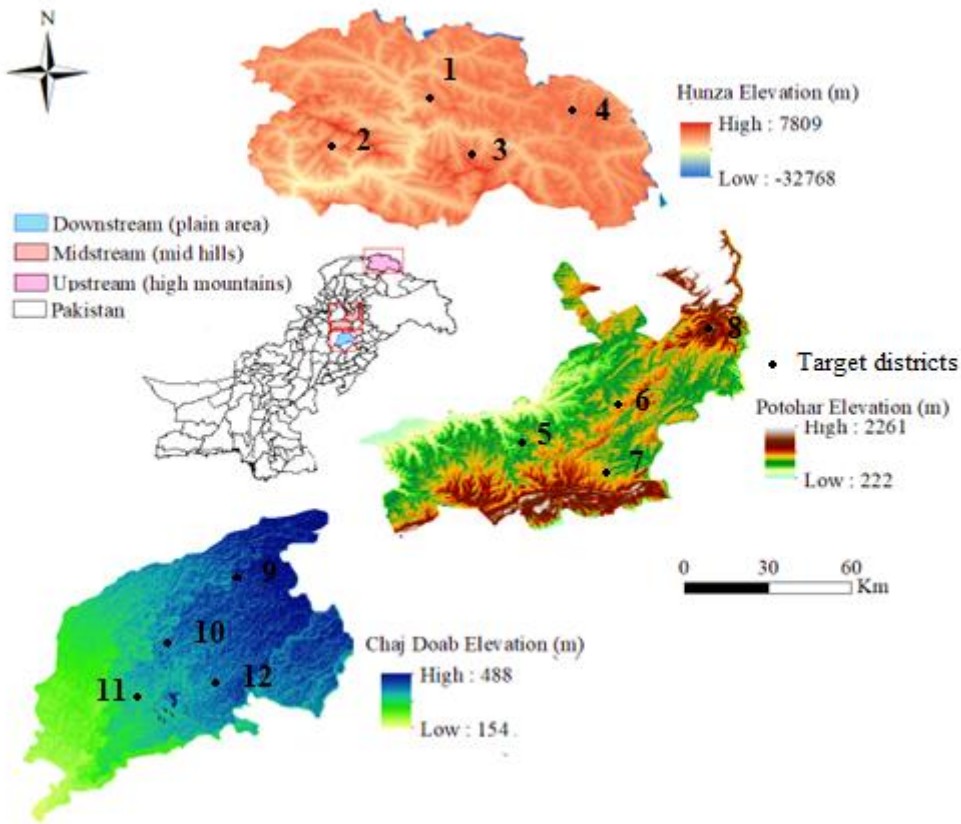

**Figure 1.** Salient features of the study area.

*2.2. Research Design*

This study combines a qualitative and a quantitative assessment of people's perceptions of climate change and response/adaptation measures. The quantitative assessment was operationalized in a survey app in which approximately 400 households participated. The qualitative assessment combined the use of a livelihood seasonal monitoring calendar and semi-structured interviews and/or focus group discussions. The quantitative assessments were conducted in a subset of the 12 study areas of the qualitative assessment and within a total of 413 participants. It was designed to provide more in-depth contextual evidence of critical moments, following the guidance provided by Groot et al. (2017) [3].

*2.3. Sampling*

For calculating the sample size of the Upper Indus Basin, Cochran's sample size formula (Cochran, 1977) [10] was used.

$$n = \frac{Z^2 \, p(1-p)}{d^2}$$

where

n = sample size (402);
p = % of households picking a choice (expressed as decimal = 0.5);
Z = Z-value (1.96 for 95% confidence interval);
D = design effect (1.50).

It is best to use simple random sampling to guarantee statistical robustness. This analysis utilized a stratified random sample approach, which could diminish the sampling procedure's statistical robustness. The design impact is essentially the ratio of the variance computed using the assumption of simple random sampling to the variance under the

actual sampling method utilized. To compensate for the loss of statistical robustness, a design effect of 1.50 has been implemented.

For each river basin, a sample size of 402 homes was established using [10], if each basin had a population of more than 50,000. Nonetheless, in all river basins, the actual number of surveyed households is more than the estimated sample sizes (Table 1), and their settlements are illustrated in Table 2.

**Table 1.** Sample size of the study sites.

| River Basins | Altitude | Determined Sample Size | Actual Surveyed Sample Size |
|---|---|---|---|
| Indus (Pakistan) | Upstream | 134 | 127 |
| | Midstream | 134 | 131 |
| | Downstream | 134 | 155 |
| | Total | 402 | 413 |

**Table 2.** Distribution of sample size across settlements.

| River Basins | Altitude | Target Districts | Selected Settlements within Districts | Number of Households | Distributed Sample Size | Actual Surveyed Households |
|---|---|---|---|---|---|---|
| Indus (Pakistan) | Upstream | Nagar | Hopper | 800 | 110 | 90 * |
| | | Hunza | Passu | 115 | 16 | 18 |
| | | | Gircha | 60 | 9 | 19 |
| | Midstream | Chakwal | Akwaal | 200 | 29 | 29 |
| | | Rawalpindi | Saroba | 350 | 52 | 38 * |
| | | | Dhok Chawan | 100 | 15 | 23 |
| | | | Gang | 250 | 37 | 41 |
| | Downstream | Sargodha | Chak 7 | 750 | 65 | 81 |
| | | | Sada Kamboh | 800 | 69 | 74 |

Note: * In five settlements, fewer households have been surveyed than the anticipated subsamples. The number of polled households represents at least 10% of all the households in these settlements; hence, survey sub-sample sizes are still statistically reliable.

A stratified sampling strategy was used to survey the individuals who were sampled. Three strata—upstream, midstream, and downstream—were developed for each river basin, such as Indus, Ganges, Gandaki, and Teesta—in light of the noteworthy variations in socioeconomic, climatic, and biophysical aspects. Due to the significant differences in household population between strata, equal sub-samples were assigned to each stratum within river basins (Table 1) to prevent inaccurate stratum-specific results from being obtained from the small sample size. Districts and study communities within districts were purposefully chosen in each stratum due to their high vulnerability to climate change-related effects. Using the "probability proportional to size (PPS)" method, the sub-sample of each stratum was dispersed throughout several chosen communities (Table 2). The number of houses in each settlement needed to perform the survey using a random route was estimated before data collection. The researchers were unable to obtain a list of families and their serial numbers in several locations, notably in mountainous regions. In these circumstances, random route sampling is a suitable technique for carrying out a field survey.

Due to the purposeful selection of districts and settlements within districts, this study's data and findings may not be true representatives of the river basin level and different altitudes, such as upstream, midstream, and downstream, which restricts the scope of generalizing study findings at river basin and altitude levels. As evidenced by the headwaters, midstream, and downstream regions of river basins, these studies should be viewed as quantitative case studies.

*2.4. Data Collection Tools*

A detailed questionnaire was created to gather information from individuals. The survey was converted to digital form using the "Akvo-flow" app for smartphones and tablets. In a training workshop, canvassers received instructions on how to use this application and conduct the surveys. To make the necessary corrections to ensure question consistency and efficient data collection by electronic devices, the questionnaire was pre-tested in all strata in April and May 2017. The actual survey was administered to every study site and took place between June and September 2017. Opcodes were used to choose the respondents in the households based on two factors. In order to answer many of the questionnaire's questions regarding respondents' perceptions of previous events, the respondent needs, first and foremost, to be older than 25. They also need to be able to recall the specifics of the events that occurred five to 10 years ago. Second, if both male and female household members (>25 years old) are available to ensure their representation in the sampling, it is preferable to interview female members (where they are active in agriculture, livestock, or other livelihood activities).

*2.5. Qualitative Assessment*

The large numbers of focus group discussions (FGDs) and group discussions (GDs) were carried out across the 12 study areas to be able to capture information on critical stress moments related to key livelihood patterns being followed by the communities residing in the high–mid–low elevations of the Hindukush Himalayan region.

### 3. Results

*3.1. Socioeconomic Characteristics of the Study Areas*

The study area in the Indus Basin was divided into three major sites: downstream (Chaj Doab, mainly Sargodha district); midstream (Soan Basin, mainly Rawalpindi and Chakwal districts); and upstream (Hunza and Nager Districts). It is found that these three regions are highly variable in terms of socioeconomic indicators of education, agriculture, income, women empowerment, health, access to basic resources, and livelihood diversifications. This variation is a result of geography, remoteness, socio-cultural and socio-political set-up, and exposure to hazard and climatic risks.

Most households in the study area have a patriarchal structure inherently as per socio-cultural norms. In hills, due to the out-migration of male members, large numbers of ownership lies with females. The literacy rates in upstream, midstream, and downstream are 65%, 80%, and 89%, respectively. Since there are limited livelihood options except for agro-pastoral activities, more people are highly focused on education nowadays. Various ethnic groups reside in different sites, and existing socio-cultural norms play a role in governing access to various assets.

Agriculture, livestock, rent, business, formal salary or wages, and casual labor are major income sources in all basin sites, among which agriculture is the primary source of income. Pastoralism, tourism, forest products, and horticulture in Upper Indus and fisheries in plains are other prominent site-specific sources of livelihood. The midstream of Pakistan suffers from low and erratic rainfall, having a dramatic impact on agricultural productivity, livestock, water, and other key sectors. Over the past decades, changing rainfall patterns and temperature fluctuations have increased the difficulties for those engaged in agriculture and rural livelihoods—particularly subsistence farmers and landless individuals. As a result, food security and poverty are major issues, the underlying cause of which is rooted in the heavy livelihood reliance on natural resource-based sectors. Flash floods and river bank erosion also damage agricultural lands.

People in upstream areas lack basic health facilities. They have to cover 500 km to come midstream (Rawalpindi/Islamabad) for any emergency treatment, while both mid- and downstream areas have easy access to health facilities. The detailed assessment of socioeconomic features is given in Table 3.

**Table 3.** Socioeconomic features in response to climate change.

| River Basins | Indus | | |
| --- | --- | --- | --- |
| | Upper Stream | Mid-Stream | Down Stream |
| House Hold (HH) Head (%) | | | |
| Male | 98.4 | 90.2 | 84.5 |
| Female | 1.6 | 9.9 | 15.5 |
| Education status of the HH head (%) | | | |
| Illiterate | 34.9 | 19.7 | 10.6 |
| Primary | 21.4 | 34.9 | 19.9 |
| Intermediate | 9.5 | 12.1 | 22.5 |
| Secondary | 22.2 | 21.2 | 29.8 |
| Bachelor | 7.1 | 5.3 | 9.3 |
| Master | 4.8 | 6.8 | 8.0 |
| Above Master | 0.0 | 0.0 | 0.0 |
| Access to agricultural land (%) | | | |
| Yes | 68.5 | 39.39 | 38.1 |
| No | 31.5 | 60.61 | 61.9 |
| Major income sources (%) | | | |
| Agriculture and forest product | 68.5 | 43.2 | 51.6 |
| Formal salary/wages | 20.5 | 15.9 | 14.8 |
| Livestock and Fishing | 13.4 | 20.5 | 20.0 |
| Remittances | 8.7 | 6.1 | 5.8 |
| Transfer Payments or Subsidies | 7.9 | 0.8 | 5.8 |
| Rent and Business | 7.1 | 6.8 | 13.6 |
| Casual labor/Piece work | 3.9 | 14.4 | 15.5 |
| Tourism | 0.8 | - | - |

*3.2. Climate Change Perceptions*

3.2.1. Perceptions of Changes in Climatic Parameters

This study shows that a majority of the surveyed population in the Upper Indus Basin, when asked about their perception of change in climate indicators over the past 10 years (2007–2017), acknowledges a change in the climate parameters, including average temperature, average rainfall, and erratic rainfall. However, the changes in average temperature were observed to be very stark in the Indus Basin, with a near absolute (97%) in the upstream and mid-stream and 94% number of respondents in the downstream, indicating a rise in average temperatures. About 96% responded to a rise in summer temperature in sub-regions, whereas the response to winter temperature is highly variable. About 63%, 38%, and 77% responded to an increase, whereas 25%, 53%, and 21% responded to a decrease in winter temperature in the up, mid, and downstream, respectively. No changes in winter temperature were reported by 11.8% and 8.4% of respondents in the upstream and downstream regions. Table 4 describes the climatic indicators of in situ people.

**Table 4.** Indicators of climate change**.**

| River Basins | Indus | | |
| --- | --- | --- | --- |
| | Upper Stream | Mid-Stream | Down Stream |
| Average Temperature (%) | | | |
| Increased | 96.9 | 97.0 | 93.6 |
| Decreased | 0.0 | 0.8 | 0 |
| No Change | 3.2 | 2.3 | 6.5 |
| Not Applicable | 0.0 | 0.0 | 0 |
| Temperature in Summer (%) | | | |

| | | | |
|---|---|---|---|
| Increased | 96.1 | 95.4 | 96.8 |
| Decreased | 0.0 | 0.0 | 1.94 |
| No Change | 3.9 | 3.8 | 1.3 |
| Not Applicable | 0.0 | 0.8 | 0 |
| **Temperature in Winter (%)** | | | |
| Increased | 63.0 | 38.2 | 77.4 |
| Decreased | 25.2 | 53.4 | 21.29 |
| No Change | 11.8 | 8.4 | 1.3 |
| Not Applicable | 0.0 | 0.0 | 0 |
| **Average Rainfall (%)** | | | |
| Increased | 44.1 | 16.0 | 51.0 |
| Decreased | 33.9 | 73.3 | 33.55 |
| No Change | 22.1 | 9.9 | 14.8 |
| Not Applicable | 0.0 | 0.8 | 0.65 |
| **Snowfall Patterns (%)** | | | |
| Increased | 10.2 | 4.6 | 9.0 |
| Decreased | 77.2 | 16.0 | 3.87 |
| No Change | 11.8 | 3.1 | 14.2 |
| Not Applicable | 0.8 | 76.3 | 72.9 |
| **Number of Dry Days (%)** | | | |
| Increased | 56.7 | 85.5 | 67.1 |
| Decreased | 7.1 | 1.5 | 1.29 |
| No Change | 29.1 | 9.2 | 17.4 |
| Not Applicable | 7.1 | 3.8 | 14.19 |

The increase in average rainfall reported was highest (51%) downstream, 44% upstream, and only 16% midstream. About 73% of midstream and 33% of up- and downstream respondents reported a decrease in average rainfall, whereas 14% of respondents felt no change compared to a decade earlier. A plausible result of this decrease in average rainfall in the midstream is an increase in the number of perceived dry days, with 85.5% of the respondents identifying an increase, potentially implying a greater risk to water availability and access for communities in this elevation. However, the number of dry days was perceived to be increasing on average in all sub-basins, with 85.5% of midstream, 67.1% of downstream, and 56.7% of the upstream respondents implying a shortening of the monsoon months. About 29.1% and 17.4% of respondents perceived no changes in dry days in the upstream and downstream regions. Moreover, snowfall patterns in the Upper Indus Basin are perceived to have been decreasing, as reported by 77% of respondents, contrary to an increase of 10.2%.

### 3.2.2. Perceptions of Extreme Weather Events

A large portion of households perceived a change in the incidence of extreme events such as floods, droughts, extreme rainfall, landslides, and heat waves. However, the nature, frequency, and intensity of extreme events differed across the three regions. In the upstream regions of the Indus Basin, our data suggest that the greatest perceived change is in flood events. The important aspects of people's perceptions of extreme events are illustrated in Table 5. However, this is in contrast to our data, which identified a rather unique and emerging threat of heatwaves in both the midstream and downstream regions of the Indus Basin. About 80% feel an increase in the frequency and intensity of heat waves in Sargodha areas.

**Table 5.** Households' perceptions of natural hazards or extreme events attributed to climate change.

| | Perceived Change in Hazards/Events | | | * Increase in Frequency of Hazards/Events | | | * Intensity of Hazards/Events | | |
|---|---|---|---|---|---|---|---|---|---|
| | US | MS | DS | US | MS | DS | US | MS | DS |
| Flood | 33.1 | 6.1 | 38.7 | 76.2 | 25.0 | 40.0 | 81.0 | 50.0 | 78.8 |
| Extreme Rainfall | 27.6 | 22.0 | 27.1 | 88.6 | 69.0 | 52.4 | 80.0 | 86.2 | 73.8 |
| Landslide | 26.0 | 3.8 | | 66.7 | 60.0 | | 78.8 | 80.0 | |
| Water Scarcity | 18.2 | 4.2 | 0.6 | 55 | 15 | 25 | | | |
| Cold Waves | 13.4 | 2.3 | | 64.7 | | | 88.2 | | |
| Erosion | 8.7 | | 3.2 | 90.9 | | 40.0 | 90.9 | | 40.0 |
| Pest Attack on Diseases | 6.3 | 7.6 | 3.2 | 87.5 | 10.0 | 100.0 | 87.5 | 10.0 | 80.0 |
| Drought | 5.5 | 26.5 | 1.9 | 42.9 | 71.4 | 33.3 | 57.1 | 74.3 | 33.3 |
| Heat Waves | 5.5 | 39.4 | 42.6 | 71.4 | 53.9 | 83.3 | 57.1 | 53.9 | 61.7 |
| Glacial Lake Outburst | 3.9 | | | 40.0 | | | 40.0 | | |
| Thunder Storm | 3.2 | | 0.7 | 75.0 | | 100.0 | 100.0 | | 100.0 |
| Cloudburst | 2.4 | | | 100.0 | | | 100.0 | | |
| Water Logging | 0.8 | 2.3 | 6.5 | | | 10.0 | | | 10.0 |
| Snow Storm | 0.8 | 4.6 | | | 33.3 | | | 16.7 | |
| Storm | 0.0 | | 1.3 | | | 50.0 | | | 50.0 |
| Hail | 0.0 | | 0.7 | | | | | | |
| Siltation | 0.0 | | | | | | | | |
| Outbreak of Diseases | 0.0 | | 2.6 | | | | | | |
| Erratic Rainfall | 0.0 | | | | | | | | |
| Forest Fire | 0.0 | | | | | | | | |
| Others | 0.0 | | | | | | | | |

* Computed among those who perceived changes in respective events. US: upstream; MS: midstream; DS: downstream.

The household perception of climate change impacts on agriculture is presented in Table 6. Agriculture is the main occupation in both midstream and downstream. More than 87% of households downstream are agriculturally dependent, while in midstream areas, 80% of households are dependent on agriculture. The major crops grown in midstream are wheat and maize. In the downstream areas, people grow wheat, sugarcane, and rice, while potatoes and wheat are mostly grown upstream.

**Table 6.** Household perception of climate change impacts on agriculture.

| River Basins | Indus | | |
|---|---|---|---|
| | Up Stream | Mid-Stream | Down Stream |
| Crop Productivity (% of responses) | | | |
| Increased | 20.5 | 12.2 | 27.1 |
| Decreased | 49.6 | 62.6 | 40.65 |
| No change | 27.6 | 16.0 | 12.9 |
| Not applicable | 2.4 | 9.2 | 19.35 |
| Incidence of crop pests (% of responses) | | | |
| Increased | 78.7 | 74.1 | 56.1 |
| Decreased | 4.7 | 1.5 | 7.74 |
| No change | 16.5 | 12.2 | 9.7 |
| Not applicable | 0.0 | 12.2 | 26.45 |
| Livestock productivity (% of responses) | | | |
| Increased | 13.4 | 10.7 | 31.0 |
| Decreased | 37.8 | 52.7 | 35.48 |

| | | | |
|---|---|---|---|
| No change | 44.9 | 29.8 | 15.5 |
| Not applicable | 3.9 | 6.9 | 18.06 |
| Incidence of livestock diseases (% of responses) | | | |
| Increased | 78.7 | 58.8 | 68.4 |
| Decreased | 4.7 | 3.8 | 7.74 |
| No change | 16.5 | 23.7 | 5.2 |
| Not applicable | 0.0 | 13.7 | 18.71 |
| Fish production (% of responses) | | | |
| Increased | 3.9 | 2.3 | 14.2 |
| Decreased | 7.1 | 4.6 | 13.55 |
| No change | 28.4 | 11.5 | 7.1 |
| Not applicable | 60.6 | 81.7 | 65.16 |
| Degradation of rangelands/pastures | | | |
| Increased | 78.0 | 23.7 | 31.6 |
| Decreased | 4.7 | 4.6 | 10.32 |
| No change | 15.0 | 32.8 | 18.1 |
| Not applicable | 2.4 | 38.9 | 40 |

## 4. Discussion

This study represents the first comprehensive analytical effort to evaluate the knowledge and views of Pakistan's vulnerable communities regarding changing climate and to investigate the gaps between locals' views and local scientists' observations. Utilizing both quantitative and qualitative evaluations, information was collected in the Upper Indus Basin at various altitudes (upstream, midstream, and downstream). The findings offer crucial insights into what people think and believe based on their personal experiences [24]. Our results are in line with earlier studies that highlight the need to integrate the assessment of hydroclimatic observational records and public perceptions in assessing the impact of climate change [14,25,26]. The result of previous studies [3,27] concluded that the current socioeconomic and socio-cultural conditions mean that percentage values of indicators like education, agriculture, annual income, access to services, and livelihood diversification are highly variable across the different elevations [28], depending on the geography, remoteness, socio-cultural and sociopolitical set-up, and exposure to hazard [29] and climatic risks, which is completely parallel with the findings of our study. In mountainous areas, households are dependent on agriculture and livestock for their livelihood [30]. The literacy rate in upstream is significantly higher compared to midstream and downstream areas.

The changing climate is hugely affecting the socioeconomic features of living inhabitants [10]. Table 3 illustrates the socioeconomic features of all local people. In all basin sites, the main sources of earnings include agriculture, livestock, rent, business, formal salary or wages, and informal services. Other notable site-specific sources of income include pastoralism, tourism, horticulture, and forest products in the Upper Indus, as well as fishing in the plains. Low and irregular rainfall in the middle of Pakistan has a significant negative influence on cattle, water, and other important industries. Changes in temperature and rainfall patterns over the past few decades have made it more challenging for people who depend on agriculture and rural livelihoods, especially landless people and subsistence farmers [31,32]. Furthermore, in comparison to mid and downstream families, upstream households have more access to fundamental government services. Because of the high literacy rate, women's empowerment has been determined to be significantly higher upstream than downstream. The previously published articles concluded similar outcomes based on people's perceptions in different parts of the world [1,13,15,33–35].

People from all basins (upstream, midstream, and downstream) observed a significant increase in temperature [5]. About 96% responded to a rise in summer temperature

in sub-regions, whereas the response to winter temperature is highly variable. Due to less rainfall throughout all sub-basins, the number of dry days was rising [36].

The in situ inhabitants perceive variations in the factors that influence the climate, such as average temperature, average precipitation, and unpredictable rainfall. However, it was found that increases in average temperature were particularly pronounced in the Indus Basin, with nearly all respondents (97%) in the upstream and mid-stream and 94% in the downstream saying that average temperatures were rising. Upstream, high rainfall and unpredictable snowfall have changed their cropping patterns and, hence, affected productivity. In the midstream section of the Indus Basin, drought and severe rainfall dangers are increasing both in frequency and intensity, whereas extreme rainfall and landslides after floods in the Upper Indus Basin, floods, and extreme rainfall following heatwaves downstream are the second and third rising hazards. Moreover, the increase in temperature and the downfall of rainfall negatively affect the crop yield in all sub-basins of the Upper Indus, causing losses in agriculture and a rise in food shortages. The inhabitants of the Indus Basin perceive the climate changing around them and report impacts of this change as increasing overall temperatures (↑US 96.9%, ↑MS 97%, ↑DS 93.6%) and erratic rainfall patterns (↑US 44.1%, ↓MS 73.3%, ↑DS 51.0%), resulting in increased water availability for crops (↑US 38.6%, ↑MS 39.7%, ↑DS 54.8%) and also increasing the number of dry days (↑US 56.7%, ↑MS 85.5%, ↑DS 67.1%) and receding crop productivity (↓US 49.6%, ↓MS 62.6%, ↓DS 40.65%).

Agriculture, being predominantly the main source of livelihood for communities across these elevations (US 68.5%, MS 43.2%, DS 51.6%), was reported to be highly vulnerable to climate change impacts and associated hazards. Major crops in all of these elevations (US Potato 44.9%, MS Wheat 77.6%, and Sugarcane 56.8%) were reported to be under strong effects of climatic hazards during the entire crop cycle; however, changes in temperature patterns (45.5%) and extreme rainfalls (45.5%) affected the harvesting (68.18%) of potato crop the most in upstream Indus areas. The best adaptive strategy reported was the introduction of new crop varieties (54.4%). Changes in temperature (46.2%) and rainfall patterns (38.5%) affected land preparation (57.69%) of wheat crops the most in midstream Indus, while changing the cropping patterns (48.3%) was the highest reported adaptive strategy. Crop growth of sugarcane (75%) was reported to be strongly affected by floods (37.5%) and changes in the rainfall patterns (25%) downstream of the Indus Basin.

The findings from this study are foundation stones in building information sources to understand the actual impact of climate change observed by people living in the Indus Basin. Moreover, being aware of public concerns could assist policymakers in creating and putting into action sustainable and successful adaptation strategies. The National Prevention Program on Climate Change may be more successful if there is widespread knowledge of the connections between climate change and human health [37]. However, there are some limitations to this study. First, because this study was only performed in three Upper Indus areas, it might not accurately reflect perceptions throughout Pakistan or the Upper Indus. Second, it is challenging to quantify and understand the subject of the investigation. This study was not able to compare public views of deteriorating health to actual health statistics or potential climatic changes. In the future, this information might be gathered to improve the precision of people's perceptions.

## 5. Conclusions

While significant efforts have been devoted to combating climate change from a scientific perspective, there is a pressing need for research and policies that specifically address indigenous knowledge and perceptions. The people's perceptions of climate change and its impacts on various socioeconomic features are very important for adaptation policies and climate mitigation [38,39–46]. Both qualitative and quantitative assessments were used to collect people's perceptions of changing climate. The main conclusion of our study is that most of the people in the Indus Basin are encountering substantial changes

in the climatic conditions and witnessing their impact. These perceptions of climate variability align closely with recorded climatic data. The majority of the study respondents noted a rise in both winter and summer temperatures across all surveyed areas. An increased number of dry days is seen in all three regions, along with a decrease in overall precipitation. People particularly associated with agriculture are observing significant changes. These include a decrease in crop productivity, an increase in crop and livestock diseases, and an increase in the degradation of pastures and rangeland. These changes impact the livelihood of the local people and make them more vulnerable to climate change.

With projected increases in global temperatures and extreme climatic events, these communities are at risk of being most impacted by climatic changes that are out of their control. Given the already severe issues of natural resource degradation and poverty in the study area, assisting people with adaptation strategies is crucial. Enhancing knowledge regarding weather patterns and forecasting, climate-smart agricultural practices, water and forest/pasture management, and livelihood diversification present viable strategies for navigating climatic fluctuations.

People's experiences with climate-related exposures should be taken into consideration by policies aiming to raise public knowledge of climate change and its health implications [5,46–50]. It is becoming increasingly clear that gaining public support for local and national strategies to address climate change and its impacts requires an understanding of peoples' perspectives and concerns. Moreover, tailoring appropriate technologies to local contexts proves beneficial. Studies such as our investigation into people's perceptions serve as foundational pillars for integrating the actual lived experiences and realities of communities into scientific research and decision-making processes.

## 6. Recommendation

This study suggests policy changes to address Indus Basin climate vulnerability, particularly for agriculture-dependent communities.

- Awareness campaigns can educate communities about long-term climate impacts on water and agriculture;
- Region-specific climate-resilient agriculture practices and training programs for farmers are recommended;
- Early warning systems and risk management plans with community participation can improve preparedness;
- Water storage infrastructure projects and sustainable water management practices are crucial;
- Collaboration with upstream countries and regional cooperation for knowledge-sharing and adaptation strategies are essential.

These recommendations can empower communities to adapt and ensure agricultural sustainability.

**Author Contributions:** All authors equally contributed. All authors have read and agreed to the published version of the manuscript.

**Funding:** This work was carried out by the Himalayan Adaptation, Water and Resilience consortium under the Collaborative Adaptation Research Initiative in Africa and Asia with financial support from the UK Government's Department for International Development and the International Development Research Centre, Ottawa, Canada. The views expressed in this work are those of the creators and do not necessarily represent those of the UK Government's Department for International Development, the International Development Research Centre, Canada or its Board of Governors, and are not necessarily attributable to their organizations.

**Institutional Review Board Statement:** Not applicable.

**Informed Consent Statement:** Not applicable.

**Data Availability Statement:** Data is contained within the article.

**Conflicts of Interest:** The author declares no conflicts of interest.

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
