# Peer review of "People’s Perception of Climate Change Impacts on Subtropical Climatic Region: A Case Study of Upper Indus, Pakistan"

_climate, doi:10.3390/cli12050073_

Round 1
Reviewer 1 Report
Comments and Suggestions for Authors
- What is the main gap in your research field?
- What new and innovative features does your research have?
The goals and necessity of your paper are not clear in the introduction section
- The discussion needs a general revision.
- In the conclusion section, you should state a general conclusion of the work. Revise this part again
Author Response
Overall response:
We are pleased that our manuscript was reviewed by such a qualified reviewer. We appreciate the positive feedback from the reviewer. Considering the review comments, we have modified the entire manuscript. Moreover, together with the revised version of the manuscript, a questionnaire sheet is also included to clarify the study's additional objectives. We hope that these revisions have improved the quality of the work such that you now may deem it worthy of publication in “Climate”. Next, we offer detailed responses to your comments. All changes made in the revised version of the manuscript have been tracked (by the track change option).
|
Sr. No. |
Comments/Suggestions |
Response |
|
1 |
What is the main gap in your research field? |
Added to the introduction section |
|
2 |
What new and innovative features does your research have? |
The literature review revealed that there is currently insufficient information available re-garding how climatic patterns are preceived by the people living in the Indus Basin and the impacts they experience. Therefore, this detailed study is designed to bridge this gap and examines the preceptions of local communities of the Indus Basin regarding climate change and its associated risks. This research aims to integrate both quantitative and qualitative examination of the dynamics of socioeconomic parameters, individuals' per-ceptions of climate change [21–23], their reactions, and the actions taken to adapt to it |
|
3 |
The goals and necessity of your paper are not clear in the introduction section |
It has been added to the section |
|
4 |
The discussion needs a general revision |
The discussion section has been revised as per reviewer's suggestion. |
|
5 |
In the conclusion section, you should state a general conclusion of the work. Revise this part again |
The conclusion section has been revised |
Reviewer 2 Report
Comments and Suggestions for Authors
I have read and reviewed the following manuscript entitled "People's Perception of Climate Change Impacts in Subtropical Climatic Region: A Case Study of Upper Indus, Pakistan" and I think that this article is interesting and has the potential to make a significant contribution, especially in the field of social sciences perspective in relation with the climate change topic. However, unfortunately, this paper can not be accepted for publication in this form and a deep modification and improvement are needed.
1- This research and most of the statistical results are based on the population survey and their perception without taking into consideration their low educational level and without improving these results with basic and confident statistical data such as the average annual precipitation issued from the meteorological stations.
2- Results of the variation of the different climate change factors or indicators are mentioned without indicating the duration or the period of this variation. I think that readers should know the period of the climatic change increasing or decreasing e.g. when the authors said that the temperature in summer increased in the upstream region by 96.1%, we should know this variation is noticed during what period of time.
3- I suggest ameliorating the introduction session by highlighting the importance of the social sciences perspective on the climate change topic and by adding some sentences at the end of this session to clarify the organisation of the different parts of the manuscript.
4. The discussion session is lost
5. The conclusion is very vague and can be improved
In conclusion, despite the importance of the study, the scientific sound is dismissed to be reliable and trustworthy and has a lot of lack. So, I think that the work should be improved and well-corrected before it is resubmitted and I encourage the authors to correct the work as soon as possible to be published.
However, there are a few things authors need to address:
-Page1 line 27: The abbreviation of the Midstream is MS and not US, please check it
-Page 3 line 125: the unit of the area should be written in kilometres square or km² and not in kilometers2 ("kilometres two")
-Figure 1: you should more clarify these figures by adding the coordinates of the region (the frame of the maps) and it will be better if you add the labels that indicate the names of the different regions of the study area (the targets districts and the 12 study areas used in the qualitative assessment)
-Page 7 line 257: please check the sentence, I think is superfluous
-Page 8 form lines 285 to 300: please check the form of the text because is written adjacent to the table
-Page 9 and 10 (tables 5 and 6): please indicate the unit of the climate change indicators perception
Finally, I would like to thank the authors for their efforts and the choice of this paper's topic and my final decision is that the manuscript can be accepted after a major revision.
Comments on the Quality of English LanguageModerate editing of the English language required
Author Response
I have read and reviewed the following manuscript entitled "People's Perception of Climate Change Impacts in Subtropical Climatic Region: A Case Study of Upper Indus, Pakistan" and I think that this article is interesting and has the potential to make a significant contribution, especially in the field of social sciences perspective in relation with the climate change topic. However, unfortunately, this paper cannot be accepted for publication in this form and a deep modification and improvement are needed.
Overall response:
We are pleased that our manuscript was reviewed by such a qualified reviewer. We appreciate the positive feedback from the reviewer. Considering the review comments, we have modified the entire manuscript. Moreover, together with the revised version of the manuscript, a questionnaire sheet is also included to clarify the study's additional objectives. We hope that these revisions have improved the quality of the work such that you now may deem it worthy of publication in “Climate”. Next, we offer detailed responses to your comments. All changes made in the revised version of the manuscript have been tracked (by the track change option).
|
Sr. No. |
Comments/Suggestions |
Response |
|
1 |
This research and most of the statistical results are based on the population survey and their perception without taking into consideration their low educational level and without improving these results with basic and confident statistical data such as the average annual precipitation issued from the meteorological stations. |
Some literature about the scientific evidence regarding the climatic conditions has been added. |
|
2 |
Results of the variation of the different climate change factors or indicators are mentioned without indicating the duration or the period of this variation. I think that readers should know the period of the climatic change increasing or decreasing e.g. when the authors said that the temperature in summer increased in the upstream region by 96.1%, we should know this variation is noticed during what period of time. |
Information about time periods has been added. As this survey was done in 2017, and during the survey people was asked to base their observation based on the past 10 years. |
|
3 |
I suggest ameliorating the introduction session by highlighting the importance of the social sciences perspective on the climate change topic and by adding some sentences at the end of this session to clarify the organisation of the different parts of the manuscript. |
The Introduction section has been revised as per reviewer's suggestion. |
|
4 |
The discussion session is lost |
The discussion section has been revised as per reviewer's suggestion. |
|
5 |
The conclusion is very vague and can be improved |
Conclusion has been revised |
|
6 |
Page1 line 27: The abbreviation of the Midstream is MS and not US, please check it |
Corrected |
|
7 |
Page 3 line 125: the unit of the area should be written in kilometres square or km² and not in kilometers2 ("kilometres two") |
Corrected |
|
8 |
Figure 1: you should more clarify these figures by adding the coordinates of the region (the frame of the maps) and it will be better if you add the labels that indicate the names of the different regions of the study area (the targets districts and the 12 study areas used in the qualitative assessment) |
Thanks for the suggestion, in our research, we used 12 different study areas which have been added to the supplementary materials of the research. |
|
9 |
Page 7 line 257: please check the sentence, I think is superfluous |
The line has been corrected as per the reviewer's suggestion. |
|
10 |
Page 8 form lines 285 to 300: please check the form of the text because is written adjacent to the table |
Corrected |
|
11 |
Page 9 and 10 (tables 5 and 6): please indicate the unit of the climate change indicators perception |
Corrected |
Finally, I would like to thank the authors for their efforts and the choice of this paper's topic and my final decision is that the manuscript can be accepted after a major revision.
Reviewer 3 Report
Comments and Suggestions for Authors
Reviewer Report for the Manuscript: climate-2955495
“People's Perception of Climate Change Impacts in Subtropical Climatic Region: A Case Study of Upper Indus, Pakistan”
Journal Name: Journal of climate
REVIEWER REPORT
Summary
The manuscript subject is good and scientific community in the filed can benefit out the study results. the study tries to explores the gaps between residents' perceptions and this region's scientific observations that is a good subject for a country like Pakistan where has been suffered drastically from climate risks during last 5 years. However, the manuscript lacks some principles as follows.
Abstract
- This section lacks a comprehensive abstract format as the methodology and results have yet to be developed.
Introduction
- This section needs a compelling argument on the issue of climate change risks perceptions amongst affected community and theories explaining the issue.
- The contributions of the study to the concurrent literature is vague
Results
- The way results are presented are weak so that they make readers somehow confused since the results are not introduced well.
- The presentation of the result is unprofessional which makes hard to follow the results. Therefore, it is highly recommendable to use charts and visual techniques highlighting the main results.
- A separate section of policy recommendation would make the results more applicable.
Comments on the Quality of English Language
Moderate editing of English language required
Author Response
The manuscript subject is good and scientific community in the filed can benefit out the study results. the study tries to explores the gaps between residents' perceptions and this region's scientific observations that is a good subject for a country like Pakistan where has been suffered drastically from climate risks during last 5 years. However, the manuscript lacks some principles as follows.
Overall response:
We are pleased that our manuscript was reviewed by such a qualified reviewer. We appreciate the positive feedback from the reviewer. Considering the review comments, we have modified the entire manuscript. Moreover, together with the revised version of the manuscript, a questionnaire sheet is also included to clarify the study's additional objectives. We hope that these revisions have improved the quality of the work such that you now may deem it worthy of publication in “Climate”. Next, we offer detailed responses to your comments. All changes made in the revised version of the manuscript have been tracked (by the track change option).
|
Sr. No. |
Comments/Suggestions |
Response |
|
1 |
Abstract - This section lacks a comprehensive abstract format as the methodology and results have yet to be developed. |
The abstract section has been revised as per the reviewer's suggestion. |
|
2 |
Introduction - This section needs a compelling argument on the issue of climate change risks perceptions amongst affected community and theories explaining the issue. - The contributions of the study to the concurrent literature is vague |
The Introduction section has been revised as per the reviewer's suggestion. |
|
3 |
Results - The way results are presented are weak so that they make readers somehow confused since the results are not introduced well. - The presentation of the result is unprofessional which makes hard to follow the results. Therefore, it is highly recommendable to use charts and visual techniques highlighting the main results. - A separate section of policy recommendation would make the results more applicable. |
We value the feedback from the reviewers and agree that the charts and visuals can enhance the work. However, we also attempted to create graphs because the data and visualizations are so complex. As a result, by following the methodology of https://www.mdpi.com/2073-4441/15/7/1287 we kept ourselves in tables after the research results were presented.
Moreover, a separate section of policy recommendations has been added in the revised version of the manuscript. |
Reviewer 4 Report
Comments and Suggestions for Authors
The research article is a good scientific contribution in which a systematic description has been portrayed in the work, still some minor changes is suggested for possible publication.
Comment:
1. The article is well written, but some section needs to be improved and needs some grammatical corrections.
2. The sources of all the data set with year should be properly mentioned.
3. The novelty of the work needs to be clearly mentioned.
4. The objectives may be mentioned point wise.
5. A methodological chart may be incorporated in the work for better understanding.
6. More graphs and/or maps (Figures) must be included to make work more understandable.
7. Some of the recent citation must be incorporated.
8. In case of the provided maps the software names (like, Arc GIS and Q GIS with version) need to be mentioned in the methodology section.
9. Some statistics about the earlier records may be compare with the current set of results and the related articles must be cited in the concerned section. Some recent citation of the literature may be incorporated to improve the quality of the work. You can find some of the mention works mentioned below:
https://doi.org/10.1016/j.jclepro.2020.124764
https://doi.org/10.1016/j.marpolbul.2024.116089
https://doi.org/10.1016/j.rsma.2019.100583
https://doi.org/10.1016/j.rsase.2023.100943
https://link.springer.com/article/10.1007/s40808-015-0044-z
https://doi.org/10.1007/s10584-016-1769-z
10. Check and correct the line 214, 301, 342-358.
11. The line should be 3.1 Socioeconomic characteristics of the study areas.
12. The incorporation of SWOC analysis (Strength, Weakness, Opportunity, Challenge) of the nature of impact of climate changes may improve the current study.
13. There is immense scope of result and discussion section.
14. The authors need to be mentioned some recommendations based on the study.
15. There is a scope to improve the conclusion section concentrating on key highlights.
Comments on the Quality of English LanguageThere is a scope of improvement.
Author Response
Reviwer-4:
The research article is a good scientific contribution in which a systematic description has been portrayed in the work, still some minor changes is suggested for possible publication.
Overall response:
We are pleased that our manuscript was reviewed by such a qualified reviewer. We appreciate the positive feedback from the reviewer. Considering the review comments, we have modified the entire manuscript. Moreover, together with the revised version of the manuscript, a questionnaire sheet is also included to clarify the study's additional objectives. We hope that these revisions have improved the quality of the work such that you now may deem it worthy of publication in “Climate”. Next, we offer detailed responses to your comments. All changes made in the revised version of the manuscript have been tracked (by the track change option).
|
Sr. No. |
Comments/Suggestions |
Response |
|
1 |
The article is well written, but some section needs to be improved and needs some grammatical corrections. |
Thank you for acknowledging the quality of the writing. We have addressed the sections needing improvement based on your specific comments |
|
2 |
The sources of all the data set with the year should be properly mentioned. |
corrected |
|
3 |
The novelty of the work needs to be clearly mentioned. |
The literature review revealed that there is currently insufficient information available re-garding how climatic patterns are preceived by the people living in the Indus Basin and the impacts they experience. Therefore, this detailed study is designed to bridge this gap and examines the preceptions of local communities of the Indus Basin regarding climate change and its associated risks. This research aims to integrate both quantitative and qualitative examination of the dynamics of socioeconomic parameters, individuals' per-ceptions of climate change [21–23], their reactions, and the actions taken to adapt to it |
|
4 |
The objectives may be mentioned point wise. |
The introduction section improved including the updating of the objectives. |
|
5 |
A methodological chart may be incorporated in the work for better understanding. |
We tried to improve the methodology description in the revised version of the manuscript. |
|
6 |
More graphs and/or maps (Figures) must be included to make work more understandable. |
We value the feedback from the reviewers and agree that the charts and visuals can enhance the work. However, we also attempted to create graphs because the data and visualizations are so complex. As a result, by following the methodology of https://www.mdpi.com/2073-4441/15/7/1287 we kept ourselves in tables after the research results were presented |
|
7 |
Some of the recent citation must be incorporated. |
corrected |
|
8 |
In case of the provided maps the software names (like, Arc GIS and Q GIS with version) need to be mentioned in the methodology section. |
The maps of software have been added in the revised version. |
|
9 |
Some statistics about the earlier records may be compare with the current set of results and the related articles must be cited in the concerned section. Some recent citation of the literature may be incorporated to improve the quality of the work. You can find some of the mention works mentioned below:
https://doi.org/10.1016/j.jclepro.2020.124764 https://doi.org/10.1016/j.marpolbul.2024.116089 https://doi.org/10.1016/j.rsma.2019.100583 https://doi.org/10.1016/j.rsase.2023.100943 https://link.springer.com/article/10.1007/s40808-015-0044-z https://doi.org/10.1007/s10584-016-1769-z |
corrected |
|
10 |
Check and correct the line 214, 301, 342-358. |
corrected |
|
11 |
The line should be 3.1 Socioeconomic characteristics of the study areas. |
corrected |
|
12 |
The incorporation of SWOC analysis (Strength, Weakness, Opportunity, Challenge) of the nature of impact of climate changes may improve the current study. |
corrected |
|
13 |
There is immense scope of result and discussion section. |
The discussion section has been revised as per the reviewer's suggestion. |
|
14 |
The authors need to be mentioned some recommendations based on the study. |
a separate section of policy recommendations has been added in the revised version of the manuscript. |
|
15 |
There is a scope to improve the conclusion section concentrating on key highlights. |
The conclusion section has been revised as per the reviewer's suggestion. |
Reviewer 5 Report
Comments and Suggestions for Authors
The manuscript is very interesting and orignal for some particular aspects of the perception of the climate change from selected environment and populations that support themselves with agricultural income. The research shows much equilibrium in the main stream question of the so-called climatic change. The bibliography is abundant and not particularly useful for the reserarch.
Author Response
Overall response:
We are pleased that our manuscript was reviewed by such a qualified reviewer. We appreciate the positive feedback from the reviewer. Considering the review comments, we have modified the entire manuscript. Moreover, together with the revised version of the manuscript, a questionnaire sheet is also included to clarify the study's additional objectives. We hope that these revisions have improved the quality of the work such that you now may deem it worthy of publication in “Climate”. Next, we offer detailed responses to your comments. All changes made in the revised version of the manuscript have been tracked (by the track change option).
Round 2
Reviewer 1 Report
Comments and Suggestions for Authors
The comments are done correctly
Reviewer 2 Report
Comments and Suggestions for Authors
I have re-read and revised the revised version of the following manuscript entitled "People's Perception of Climate Change Impacts in Subtropical Climatic Region: A Case Study of Upper Indus, Pakistan" and I consider that the authors have taken into consideration all comments and The recommendations are well edited and the quality of the manuscript is significantly improved and can be accepted for publication in this current form.